# PAXION: Patching Action Knowledge in Video-Language Foundation Models

**Zhenhailong Wang**[1], **Ansel Blume**[1], **Sha Li**[1], **Genglin Liu**[1],
**Jaemin Cho**[2], **Zineng Tang**[2], **Mohit Bansal**[2], **Heng Ji**[1]
[1]UIUC  [2]UNC
{wangz3,hengji}@illinois.edu

## Abstract

Action knowledge involves the understanding of textual, visual, and temporal aspects of actions. We introduce the **Action Dynamics Benchmark (ActionBench)** containing two carefully designed probing tasks: Action Antonym and Video Reversal, which targets multimodal alignment capabilities and temporal understanding skills of the model, respectively. Despite recent video-language models' (VidLM) impressive performance on various benchmark tasks, our diagnostic tasks reveal their surprising deficiency (near-random performance) in action knowledge, suggesting that current models rely on object recognition abilities as a shortcut for action understanding. To remedy this, we propose a novel framework, **PAXION**, along with a new **Discriminative Video Dynamics Modeling (DVDM)** objective. The PAXION framework utilizes a **Knowledge Patcher** network to encode new action knowledge and a **Knowledge Fuser** component to integrate the Patcher into frozen VidLMs without compromising their existing capabilities. Due to limitations of the widely-used Video-Text Contrastive (VTC) loss for learning action knowledge, we introduce the DVDM objective to train the Knowledge Patcher. DVDM forces the model to encode the correlation between the action text and the correct ordering of video frames. Our extensive analyses show that PAXION and DVDM together effectively fill the gap in action knowledge understanding (~50% → 80%), while maintaining or improving performance on a wide spectrum of both object- and action-centric downstream tasks. The code and data will be made publicly available for research purposes at https://github.com/MikeWangWZHL/Paxion.git.

## 1 Introduction

Recent video-language models (VidLMs) [30, 25, 55, 35, 57, 52] have shown impressive performance on a wide range of video-language tasks. However, such multimodal models are not without deficiencies: [24] points out that many popular video-language benchmarks [56, 3, 16] can be solved by looking at a single frame, and [59] shows that vision-language models struggle to understand compositional and order relations in images, treating images as bags of objects. Such limitations suggest that models' understanding of *actions*, which may require several frames and comprehension of object relationships, may be lacking.

To test this hypothesis, we first define **action knowledge** as an understanding of the cause and effect of actions in textual, visual, and temporal dimensions. To quantify a model's action knowledge, we introduce the **Action Dynamics Benchmark (ActionBench)**. ActionBench contains two probing tasks: distinguishing between (1) a video's caption and the caption with its action verbs replaced by their antonyms; (2) the original and reversed videos. The benchmark also includes a baseline task for controlling the undesired impact from domain mismatch and investigating potential bias towards objects. The baseline task requires the model to differentiate between the original video captions

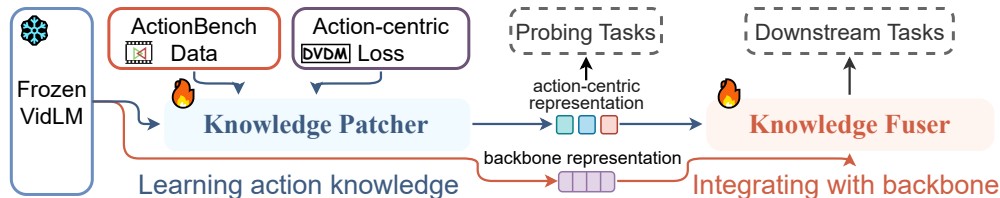

Figure 1: Overview of the PAXION framework. The goal is to patch frozen VidLMs with action knowledge without compromising their general vision-language capabilities. The Knowledge Patcher (KP) aims to learn an action-centric representation by leveraging ActionBench data (§ 2) and our newly proposed Discriminative Video Dynamics Modeling (DVDM) training objectives (§ 3.1). The Knowledge Fuser (KF) aims to obtain a balanced representation for general downstream tasks by fusing the KP with the backbone.

and altered versions with randomly replaced objects. We find that state-of-the-art video-language foundation models [52, 57, 24] exhibit near-random performance on our action-oriented probing tasks while excelling on the object-oriented baseline task (Figure 2). This shows that VidLMs lack action knowledge and suggests that their impressive performance on other benchmarks may be attributed to their object recognition ability instead of action understanding.

To address this shortcoming, we propose a novel framework, **PAXION** (Patching Actions), to patch existing VidLMs with action knowledge without compromising their general vision-language (VL) capabilities. PAXION comprises two main components, the Knowledge Patcher and the Knowledge Fuser. The **Knowledge Patcher (KP)** is a Perceiver-based [21, 20] lightweight module attached to a frozen VidLM backbone used to augment the VidLM with action-aware representations. Through our preliminary experiments, one major challenge for patching action knowledge is that the widely-used Video-Text Contrastive (VTC) objective [43, 55, 29, 26, 28] is insufficient, which echoes the findings of related work [7, 24, 59]. Hence, inspired by dynamics modeling in robotics and reinforcement learning [1, 4, 15, 23, 39], we introduce the **Discriminative Video Dynamics Modeling (DVDM)** objective that forces the model to learn the correlation between an action's textual signifier, the *action text* (e.g. the word "falling"), and the action's visual depiction (e.g. a clip of a falling book). DVDM includes two new losses, *Video-Action Contrastive (VAC)* and *Action-Temporal Matching (ATM)*, which are compatible with VTC without additional parameters. Specifically, we formulate discriminative tasks using action antonyms and reversed videos, with special emphasis on learning from data instances with salient state changes. We demonstrate that the synergy between the Knowledge Patcher and DVDM leads to a dramatic improvement on our ActionBench tasks.

Next, we investigate whether our Knowledge Patcher, which is specialized for action understanding, can be integrated into existing VidLMs for downstream tasks that require both action and object knowledge. To this end, we introduce the **Knowledge Fuser (KF)** component of PAXION which fuses the *action-centric representation* from the Knowledge Patcher with the *object-centric representation* from the frozen backbone using cross-attention. We show that the fused representation from PAXION improves both object and action understanding on a wide spectrum of tasks, including Video-Text Retrieval (SSv2-label [13, 24]), Video-to-Action Retrieval (SSv2-template [24], Temporal [45]), and Causal-Temporal Video Question Answering (NExT-QA [54]). Moreover, our analysis shows that the Knowledge Fuser is essential to maintain a balance between the models' object-related understanding and improving performance on downstream action and temporal-oriented tasks.

We also test the robustness of PAXION by considering a zero-shot cross-domain transfer setting on the Moments-in-Time [38] and Kinetics [22] datasets. We find that the Knowledge Fuser is critical for increasing robustness to domain shifts and that positive transfer to unseen domains can be achieved by further ensembling PAXION with the backbone model.

To the best of our knowledge, this is the first work to systematically evaluate action knowledge and patch it into video-language foundation models. Our main contributions are threefold:

1. We introduce the Action Dynamics Benchmark (§ 2), which probes action understanding capabilities in video-language models. We evaluate three state-of-the-art video-language foundation models and conclude that they lack a basic grasp of action knowledge.

2. We propose a novel learning framework called PAXION, which patches the missing action knowledge into frozen video-language foundation models without hurting their gen-

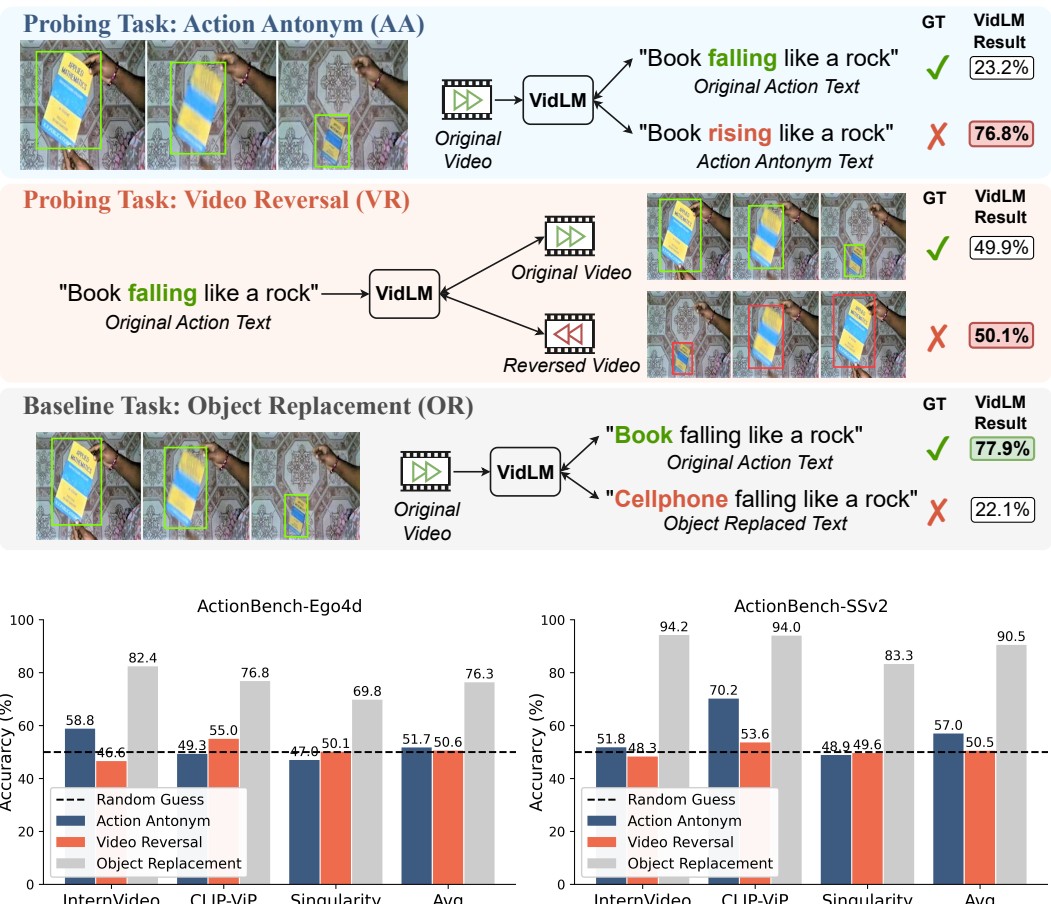

Figure 2: **Top**: Illustration of the probing tasks and baseline task in our proposed **ActionBench**. The bounding boxes in the video frames are purely for visualization. The numbers on the right show the ranking scores from a state-of-the-art VidLM, InternVideo [52]. The model struggles to determine whether a book is "falling" or "rising," but can confidently identify the object to be a "book" instead of a "cellphone". **Bottom**: ActionBench results of three recent VidLMs [52, 57, 24]. The column "Avg" indicates averaged results on each task across all three models. Existing VidLMs achieve near-random results on the probing tasks (AA and VR) while excelling on the baseline task (OR). This demonstrates that *existing VidLMs lack fundamental action knowledge and exhibit strong bias to object understanding.*

       eral vision-language capabilities. The key components of PAXION include a Perceiver-based Knowledge Patcher (§ 3) and a cross-attention-based Knowledge Fuser (§ 4).

3. We propose the DVDM objective (§ 3.1), an improvement over the widely-used VTC loss, which forces the model to encode the correlation between the action text and the correct ordering of video frames. Extensive experiments show that PAXION with DVDM improves the joint understanding of objects and actions while being robust to domain shift.

# 2 Action Dynamics Benchmark (ActionBench): Do Video-Language Foundation Models Understand Action Knowledge?

To investigate the presence of action knowledge in state-of-the-art video-language foundation models, we propose the **Action Dynamics Benchmark (ActionBench)**. ActionBench comprises the **Action Antonym (AA)** and **Video Reversal (VR)** probing tasks, along with the **Object Replacement (OR)** baseline task. The probing tasks evaluate the *multimodal and temporal correlations between an action text and a video*. The baseline task controls for the potential impact of domain mismatch.

We construct this benchmark by leveraging two existing open-domain video-language datasets, Ego4D [14] and Something-Something v2 (SSv2) [13], which provide fine-grained action annotations for each video clip. Compared to a previous verb understanding probing benchmark [42] based on MSRVTT [56] and LSMDC [44], ActionBench is more action-oriented, larger in scale, and contains both ego-centric and third-person videos. Detailed statistics can be found in Appendix B. An illustration of each ActionBench task can be found in Figure 2.

**Probing Task: Action Antonym (AA).** The Action Antonym task probes the multimodal alignment of the action text and the video representation. We formulate AA as a binary classification task that involves distinguishing the original text annotation from its altered version with the action replaced by its antonym, given the corresponding video clip. For example, if the original text is ''Book `falling` `like a rock`'', the action antonym text would be ''Book `rising` `like a rock`''. We leverage the WordNet [36] database and manually constructed mappings to automatically construct the antonym texts (details in Appendix B).

**Probing Task: Video Reversal (VR).** The Video Reversal task probes the temporal understanding of actions. We formulate VR as a binary classification task, where given a video-text pair with at least one action and a reversed version of the video, the goal is to distinguish the original video from the reversed one. Achieving non-trivial performance on the Video Reversal task requires the model to understand the temporal sequence implied by the action. The Video Reversal task also evaluates VLMs' abilities to identify violations of physical knowledge, as some clips defy expectation when reversed (e.g. a falling book becomes one which rises without any discernible cause).

**Baseline Task: Object Replacement (OR).** Object Replacement is a binary classification task that requires the model to distinguish between the original text annotation and an altered version with objects tokens randomly replaced by other object tokens in the dataset. The Object Replacement task allows us to understand: (1) whether current VidLMs rely on object recognition as a "shortcut" for video-text matching (i.e., if they have an object-biased representation), and (2) whether poor performance on Action Antonym can be attributed to domain mismatch (i.e., not being trained on Ego4D or SSv2) instead of a lack of action knowledge.

## 2.1 Evaluating Video-Language Models on ActionBench

We evaluate three recent video-language foundation models, **InternVideo** [52], **CLIP-ViP** [57] and **Singularity-temporal** [24][1], on ActionBench. Despite their impressive improvements on video-language benchmarks, these models struggle to achieve non-trivial performance on Action Antonym and Video Reversal, as depicted in Figure 2. The fact that they achieve significantly better performance on the Object Replacement task indicates a strong bias towards objects over actions, and affirms that the poor performance on AA is not solely a result of domain mismatch. The near-random performance on the VR task indicates a lack of basic temporal reasoning and physical knowledge.

These observations align with previous approaches [17, 42, 59, 37] which show similar limitations in image-language models [43, 28] and earlier video-language models [11, 35]. We find that high performance on video-language benchmarks does not necessarily equate to a stronger understanding of action knowledge.

## 3 Patching Action Knowledge in Frozen Video-Language Models

In § 2, we showed that current VidLMs exhibit limitations in their understanding of action knowledge, a crucial component for developing a comprehensive understanding of the external world. This raises the important question: *Can we enhance existing VidLMs with this missing knowledge without hurting their general video-language capabilities?*

To this end, we propose a novel learning framework, PAXION, which comprises two main components: the **Knowledge Patcher (KP)** (§ 3) and the **Knowledge Fuser (KF)** (§ 4). An overview of the PAXION framework can be found in Figure 1. Analogous to releasing *patches* to fix bugs in published software, the Knowledge Patcher is a Perceiver-based [21, 20] lightweight module attached to a frozen VidLM for steering the VidLM towards action-centric representations. As the widely used video-language contrastive (VTC) objective is insufficient for learning action knowledge, we introduce

---

[1]For simplicity, we use "Singularity" to represent "Singularity-temporal" in our figures and tables.

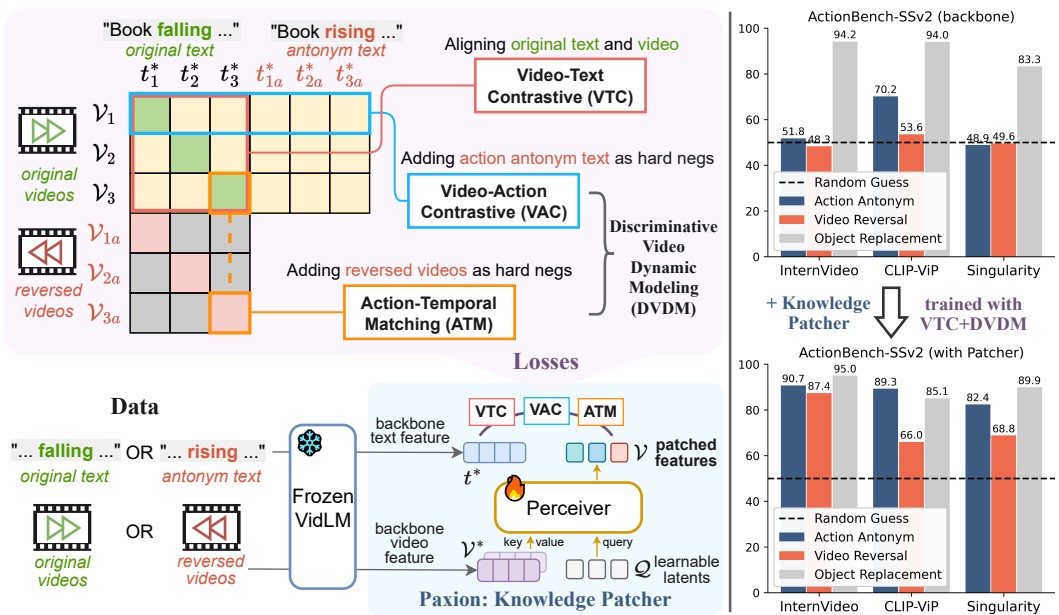

Figure 3: Illustration of the **Knowledge Patcher** component (bottom left) of PAXION and the training objectives (upper left). On the right, we show the comparison of performance on ActionBench before and after adding the Knowledge Patcher.

Table 1: ActionBench results (in accuracy %). *KP-\** refers to Knowledge Patcher. *AA* and *VR* indicate the Action Antonym task and the Video Reversal task. *VTC* and *DVDM* stands for Video-Text Contrastive loss and our newly proposed Discriminative Video Dynamics Modeling losses detailed in § 3.1. *Trainable Param#* indicates the size of the trainable parameters compared to the backbone.

| Action Dynamics Benchmark (ActionBench) Results | | | | | | | |
|---|---|---|---|---|---|---|---|
| **Backbone** | **Method** [Patcher Training Loss] | **Trainable Param#** | **AA** (Ego4d) | **VR** (Ego4d) | **AA** (SSv2) | **VR** (SSv2) | **Avg** |
| InternVideo | Backbone | - | 58.8 | 46.2 | 51.8 | 48.3 | 51.3 |
| | KP-Transformer [VTC] | 8.4M (1.8%) | 68.2 | 62.8 | 65.5 | 60.6 | 64.3 |
| | KP-Perceiver [VTC] | 4.2M (0.9%) | 66.5 | 63.6 | 69.8 | 71.0 | 67.7 |
| | KP-Perceiver [VTC+**DVDM**] | 4.2M (0.9%) | **90.1** | **75.5** | **90.7** | **87.4** | **85.9** |
| Clip-ViP | Backbone | - | 49.3 | 55.0 | 70.2 | 53.6 | 57.0 |
| | KP-Transformer [VTC] | 3.9M (2.6%) | 61.9 | 53.4 | 72.2 | 54.3 | 60.5 |
| | KP-Perceiver [VTC] | 2.4M (1.6%) | 61.9 | 54.6 | 71.5 | 48.8 | 59.2 |
| | KP-Perceiver [VTC+**DVDM**] | 2.4M (1.6%) | **89.3** | **56.9** | **89.3** | **66.0** | **75.4** |
| Singularity | Backbone | - | 47.0 | 50.1 | 48.9 | 49.6 | 48.9 |
| | KP-Transformer [VTC] | 3.9M (1.8%) | 61.9 | 48.2 | 63.8 | 49.5 | 55.9 |
| | KP-Perceiver [VTC] | 1.3M (0.6%) | 60.3 | 46.1 | 63.3 | 51.5 | 55.3 |
| | KP-Perceiver [VTC+**DVDM**] | 1.3M (0.6%) | **83.8** | **58.9** | **82.4** | **68.8** | **73.5** |
| Human | | | 92.0 | 78.0 | 96.0 | 90.0 | 89.0 |

**Discriminative Video Dynamics Modeling (DVDM)** (§ 3.1) objectives that force the model to encode the correlation between the actual action text (e.g., "falling") and the correct sequence of visual state-changes (i.e., video frames).

**Knowledge Patching with Perceivers.** Inspired by recent work [2, 27] leveraging Perceivers [21, 20] to extract *language-related* visual features, we use Perceivers to extract *knowledge-specific* features. As shown in Figure 3 Knowledge Patcher , we use a lightweight Perceiver which performs cross-attention between a sequence of lower-dimensional, learnable latents $\mathcal{Q}$ and the higher-dimensional visual embedding $\mathcal{V}^*$ from a frozen, pretrained VidLM backbone. To further investigate the viability of Perceivers as an alternative to Transformers [49], we include another variant of the KP where we replace the Perceiver with a standard Transformer Encoder. Table 1 shows that the

Perceiver-based KP achieves competitive or better performance compared to the Transformer variant while being 2-3 times smaller in scale. Architecture details of the KPs can be found in Appendix D.1.

**Video-Text Contrastive (VTC) is insufficient for learning action knowledge.** We initially train both variants of the Knowledge Patcher on the training set of ActionBench with only the Video-Text Contrastive (VTC) loss. VTC loss aligns the visual representation $\mathcal{V}$ from the KP with the pooled textual representation $t^*$ from the frozen backbone. Results in Table 1 show that training with the VTC loss alone provides marginal to no improvements on Action Antonym (AA) and Video Reversal (VR), particularly on smaller backbone models. This suggests the need for new training objectives (§ 3.1) for learning action knowledge.

### 3.1 Learning Action Knowledge with Discriminative Video Dynamics Modeling

To address the limitation of the VTC loss in learning action knowledge, we propose two new losses that draw inspiration from dynamics modeling in Robotics and Reinforcement Learning [1, 4, 15, 23, 39]. Specifically, in a typical Markov Decision Process (MDP) setup, *forward dynamics modeling* aims to predict the next world state $\hat{x}_{t+1}$ given the current world state $x_t$ and the action $u_t$. *Inverse dynamics modeling* aims to predict the current action $\hat{u}_t$ given the current and next world state $x_t, x_{t+1}$. Given video frame as a representation of the world states, existing work usually formulates forward dynamics modeling as a generative task [1, 39, 15], directly reconstructing the pixels or the latent embedding of the next frame. For inverse dynamics modeling, the action class is usually predicted using a dedicated classification layer [4, 23, 1]. However, our preliminary experiments show that the existing formulation cannot be directly applied in our setting due to the following **unique challenges**: (1) Real world videos are much more complex than videos in a lab setting, with constantly changing backgrounds and moving camera angles, causing a large portion of visual features to be unrelated to the main objects and actions. Furthermore, without additional annotation, it is difficult to identify the frames corresponding to the "current" and "next" states, as actions may be continuous (e.g., "walking") or repetitive (e.g., "doing push-ups") within a video. Thus, the training signal from a regression loss becomes extremely noisy. (2) Unlike previous work that has a small fixed number of action classes, we model actions as natural language phrases, making direct classification inapplicable.

To address these unique challenges, we propose a novel *"relaxed"* formulation of dynamics modeling, dubbed **Discriminative Video Dynamics Modeling (DVDM)**, which contains two losses: **Video-Action Contrastive (VAC)** and **Action-Temporal Matching (ATM)**. Both VAC and ATM can be directly incorporated into the Video-Text Contrastive (VTC) loss without any additional parameters. As illustrated in Figure 3 Losses , the VAC loss aims to encourage the model to learn the correlation between the visual observations and the actual actions. We formulate the VAC loss as adding action antonym texts as hard negatives. The ATM loss encourages the model to consider the temporal ordering of the visual observations (video frames). Instead of directly generating the next state frames, we formulate ATM as a discriminative task similar to Video Reversal in § 2, where the model distinguishes reversed videos from the original videos, alleviating the need for explicit state annotations. In order to make sure that the reversed videos are indeed distinguishable from the original ones, we further introduce a method (Appendix C) for identifying videos with salient state-changes by leveraging image-language foundation models [28]. The idea is to measure the frame-text and frame-frame similarity between the first and second half of a video. We compute ATM loss only on the videos that have salient state-changes between frames. Experimental results, as shown in Table 1 and Figure 3, indicate that **adding the DVDM objectives significantly improves the performance on both probing tasks**, suggesting that the resulting representation from the Knowledge Patcher demonstrates a stronger understanding of action knowledge.

## 4 Leveraging Patched Action Knowledge for Downstream Tasks

In § 3, we showed that the Knowledge Patcher (KP) and DVDM objectives together effectively learn action knowledge-specific representations. However, these representations are highly specialized to action understanding, which may not be optimal for general downstream tasks that require both object and action understanding. Thus, the remaining challenge is to *retain the general VL capabilities of the backbone while leveraging the newly learned action knowledge.*

One naive idea is to simply use the backbone embeddings whenever the task is less action-centric. However, it is difficult to decide when to use the backbone without prior knowledge of a given task. Further, using the backbone embeddings alone gives up the patched action knowledge that can be essential for certain downstream tasks, such as action recognition. In this section, we demonstrate that we can get the best of both worlds by fusing the action-centric representation from the Knowledge Patcher with the object-centric representation from the frozen backbone.

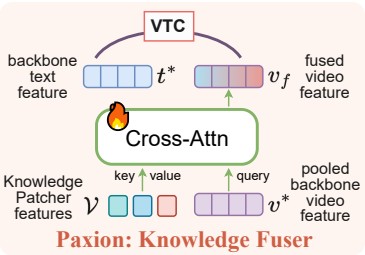

Figure 4: Illustration of the Knowledge Fuser component.

For this we introduce the second component of PAXION, the **Knowledge Fuser (KF)**, illustrated in Figure 4 . The KF takes the pooled visual feature ($v*$) from the frozen VL backbone as the input query, and performs cross-attention with the extracted visual tokens ($\mathcal{V}$) from the Knowledge Patcher.

## 4.1 Experimental Setup

To evaluate the model's ability to retain general visual-linguistic capabilities while leveraging newly learned action knowledge, we consider a spectrum of video-language tasks with different emphases on object and action understanding. Specifically, we consider **Video-Text Retrieval** (SSv2-label [24]), which is object-centric and biased towards static appearances [59, 24]; **Causal-Temporal VQA** (NExT-QA [54]), which requires joint understanding of static objects and dynamic events; and **Video-to-Action Retrieval** (SSv2-template [24], Temporal-SSv2 [45]), which is highly action-centric and temporal-intensive. A task is considered to be temporal-intensive if it cannot be solved without correct temporal information [45], e.g., reversed or shuffled frames. For example, as illustrated in Figure 5, the Video-to-Action Retrieval task obscures object names in the text, making it impossible to align text with a video based solely on objects. Moreover, it is impossible to distinguish "approaching" and "moving away" without considering the temporal ordering of the frames.

For **PAXION**, we finetune the Knowledge Fuser jointly with the Knowledge Patcher on downstream tasks using VTC loss. By default, the KP in PAXION is trained with VTC and DVDM losses (§ 3.1). We include the baselines **KP-Transformer FT** [VTC] and **KP-Perceiver FT** [VTC], which are both obtained by continuing to finetune the VTC-only KPs from Table 1 on downstream tasks. Additionally, we compare PAXION with **Side-Tuning** [61], a Parameter-Efficient Finetuning (PEFT) method that could serve as an alternative to the KF. For the Side-Tuning variant, we initialize the "side-model" using the same Knowledge Patcher as in PAXION and do alpha blending with the frozen backbone. Implementation and configuration details for each method and task can be found in Appendix D. The results are shown in Tables 2 and 3.

Table 2: Video-Text Retrieval and Video-to-Action Retrieval results. R1 and R5 represent Recall@1 and Recall@5 (in %) respectively. Subscripts $_{vt2}$ and $_{t2v}$ represent video-to-text and text-to-video, respectively.

| Method [Patcher Training Loss] | Video-Text Retrieval SSv2-label | | | | Video-to-Action Retrieval | | | |
| | | | | | SSv2-template | | Temporal-SSv2 | |
| | $R1_{v2t}$ | $R5_{v2t}$ | $R1_{t2v}$ | $R5_{t2v}$ | $R1$ | $R5$ | $R1$ | $R5$ |
| --- | --- | --- | --- | --- | --- | --- | --- | --- |
| InternVideo Backbone | 18.8 | 39.9 | 19.9 | 40.0 | 5.6 | 15.9 | 11.2 | 35.8 |
| KP-Transformer FT [VTC] | 24.1 | 50.0 | 21.7 | 46.0 | 21.1 | 55.9 | 41.1 | 88.9 |
| KP-Perceiver FT [VTC] | 27.0 | 57.4 | 27.1 | **56.8** | 24.8 | 59.7 | 42.5 | 91.3 |
| Side-Tuning [61] [VTC+DVDM] | 30.9 | 59.2 | 26.6 | 53.1 | 22.2 | 55.1 | 50.2 | 90.9 |
| **PAXION** [VTC+DVDM] | **32.3** | **61.2** | **28.0** | 54.3 | **26.9** | **61.5** | **51.2** | **91.9** |

## 4.2 Analysis

**PAXION improves joint understanding of objects and actions.** Tables 2 and 3 show that PAXION outperforms both the *Backbone* and the *VTC-only baselines (KP-\*)*. This indicates that PAXION not only retains the original VL capabilities of the backbone, but also fills in the gap of the missing action knowledge by fusing the original representations with the patched ones. We corroborate this finding by observing more significant improvements on action-centric and temporal-intensive tasks,

Table 3: Causal-Temporal VQA (NExT-QA) results (in accuracy %) on the validation set. We consider both the original and ATP-hard [7] split. We report accuracy for *all* questions or specific types of questions, including causal (*C*), temporal (*T*), and descriptive (*D*) questions.

| Method [Patcher Training Loss] | NExT-QA | | | | | | |
|---|---|---|---|---|---|---|---|
| | Original | | | | ATP-hard [7] | | |
| | C | T | D | all | C | T | all |
| InternVideo Backbone | 43.3 | 38.6 | 52.5 | 43.2 | 27.0 | 27.3 | 27.1 |
| KP-Transformer FT [VTC] | 46.1 | 45.0 | 61.3 | 48.1 | 32.5 | 33.6 | 33.0 |
| KP-Perceiver FT [VTC] | 46.0 | 46.0 | 58.9 | 48.0 | 30.1 | 31.6 | 30.7 |
| Side-Tuning [61] [VTC+DVDM] | 54.9 | 52.0 | **69.8** | 56.3 | 37.4 | 36.0 | 36.8 |
| **PAXION** [VTC+DVDM] | **56.0** | **53.0** | 68.5 | **57.0** | **38.8** | **38.1** | **38.5** |

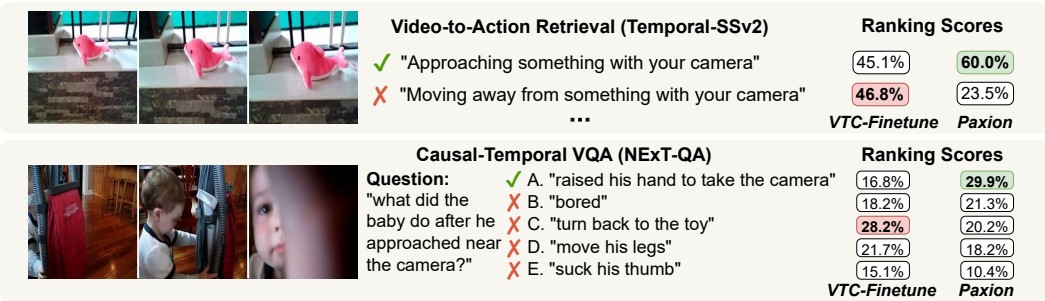

Figure 5: Qualitative examples on Temporal-SSv2 [45] and NExT-QA [54]. *VTC-Finetune* and PAXION refer to methods in row 3 and row 5 in Tables 2 and 3.

such as Temporal-SSv2 (+20% @R1), compared to object-centric tasks, such as SSv2-label (+11% @R1)[2]. The decomposed results on NExT-QA (Table 3) show that PAXION helps more on the Causal (*C*) and Temporal (*T*) types of questions, and on the harder subset (ATP-hard) where the temporal and action knowledge is emphasized.

PAXION also outperforms Side-Tuning, highlighting the effectiveness of cross-attention for deep fusion. Specifically, the Knowledge Fuser allows us to attend to all extracted visual tokens from the Knowledge Patcher instead of only blending with pooled representations as in Side-Tuning.

**Qualitative analysis.** Figure 5 shows two qualitative examples on Temporal-SSv2 and NExT-QA. For the Temporal-SSv2 example, we find that the finetuned Knowledge Patcher trained with only VTC fails to distinguish *"Moving away"* from *"Approaching,"* while PAXION trained with DVDM successfully correlates the seemingly expanding object with the action *"Approaching"*. For the NExT-QA example, the question asks the model to identify what happens after the action *"approached near the camera"*. The VTC baseline incorrectly selects the action *"turn back to the toy,"* which happens before approaching the camera. On the other hand, PAXION successfully chooses *"raised his hand to take the camera"*. This indicates a stronger understanding of both action dynamics in words such as *"approach"* and the temporal ordering implied by words such as *"after"*. Additional qualitative examples and analysis of failure cases can be found in Appendix E.

**Disentangling the impact of Knowledge Patching and Fusing.** We further investigate the disentangled impact of the Knowledge Fuser (KF) and the Knowledge Patcher (KP) with two ablation settings: (1) **KP+Finetune**, where instead of adding the KF, we directly *finetune* the KP *trained with DVDM* on downstream tasks; (2) **KP[VTC]+KF**, where we train the KP *without DVDM* and then *add the KF* upon it. The results are shown in Figure 6, where the Δ score represents the relative difference of downstream task performance between our original **PAXION** (Row 5 in Tables 2 and 3) and the two ablated settings. The key observations are as follows: (1) **The Knowledge Fuser contributes more to object understanding.** From Figure 6 Left, we find that the KF helps most when the tasks are more object-centric, e.g., SSv2-label. On highly action-centric tasks, e.g., Temporal-SSv2, directly using the action-knowledge-patched representation is preferable to fusing with the backbone representation.

---

[2]The scores are calculated between PAXION and KP-Perceiver FT [VTC]. The improvement @R1 for SSv2-label is averaged across $R1_{v2t}$ and $R1_{t2v}$.

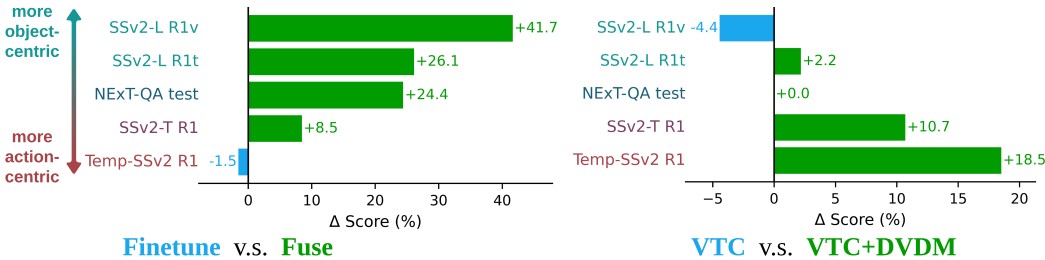

Figure 6: **Left**: Impact of the *Knowledge Fuser*. Comparing *finetuning* or *fusing* the same Knowledge Patcher trained with VTC+DVDM losses. **Right**: Impact of *action knowledge patching (DVDM)* on downstream tasks. Comparing fusing with the Knowledge Patcher trained with *VTC loss only* or *VTC+DVDM losses*. The Δ score indicates the relative difference in terms of downstream task accuracy between our original PAXION and the ablated settings (detailed in §4.2).

(2) **Patching with action knowledge contributes more to action-centric understanding.** From Figure 6 Right, we find that patching with action knowledge, i.e., training with DVDM objectives, contributes to better performance on downstream tasks that are more action-centric. Importantly, this result also indicates that the improvements observed in Tables 2 and 3 do not come solely from adding the KF. However, if the task is more object-centric, such as SSv2-label, VTC training alone is sufficient.

**Robustness to domain shift.**   Learned action knowledge should be generalizable to unseen tasks and domains. However, this goal is difficult to realize with only domain-specific datasets like SSv2[13] which contains only 174 actions. Therefore, in Appendix A we conduct experiments on zero-shot cross-domain transfer which demonstrate that the Knowledge Fuser in PAXION increases robustness to domain shift and can introduce positive transfer during zero-shot inference.

## 5   Related Work

**Limitations of vision-language contrastive pretraining.**   Since CLIP [43], multimodal contrastive losses have been the major pretraining objective for almost all recent image-language [43, 28, 27, 53, 51, 58] and video-language models [55, 35, 10, 60, 12, 50, 57, 52]. Previous work [17, 59, 24, 5] has revealed the limitation of contrastive pretraining on fine-grained compositional understanding, verb understanding, and temporal reasoning. Concurrent work [37, 9] proposed mining hard negatives by rule-based heuristics or large-language models [8] to improve understanding of structured vision-language concepts and verbs. In this work, we focus on general action knowledge which includes verb understanding as well as action temporal understanding. Instead of directly tuning the entire backbone as in [59, 37], PAXION enables fast action knowledge patching while also achieving improved performance on both object-centric and action-centric downstream tasks. It is worth noting that the hard negative mining method proposed by [37] can be easily incorporated with our VAC loss and could potentially result in stronger results.

**Parameter-efficient fine-tuning (PEFT).**   The recent surge in the size of large language models [6, 41, 62, 40, 48] has spurred research on parameter-efficient fine-tuning [18, 31, 61, 46, 19, 33]. Although current video-language models are smaller in scale, we aim to develop PAXION to be applicable to larger models that we anticipate will emerge in the near future. The most similar PEFT-related work to ours is Side-Tuning [61], which we compare against in § 4.1. At a high-level, unlike existing PEFT methods that optimize for specific downstream tasks, PAXION is designed to learn a specific type of knowledge that can benefit various downstream tasks (§ 4.2). Furthermore, it is unclear how to aggregate the task-specific parameters, such as those in adapters [18] or low-rank layers [19], to perform multiple tasks. The versatility of PAXION allows for its use in learning various types of knowledge, each with its own Knowledge Patcher. Subsequently, the patched knowledge-specific representations can be fused together using one Knowledge Fuser. This work serves a proof-of-concept where we focus on action knowledge. We leave the exploration of other types of knowledge and a more comprehensive comparison with PEFT methods as future work.

# 6 Conclusions and Future Work

In this work we propose the ActionBench benchmark for evaluating models' understanding of action knowledge, and reveal a major deficiency in state-of-the-art video-language foundation models in this area. We then propose PAXION to patch in such action knowledge without compromising models' existing capabilities. We show that PAXION significantly improves the model's action understanding while achieving competitive or superior performance on downstream tasks. One limitation of this work is that we only experimented with patching one type of knowledge. We intend to address this in future work, where we plan to expand PAXION to patch broader aspects of physical knowledge such as object affordances and mental simulation, and to explore fusion with multiple learned Knowledge Patchers.

## Acknowledgements

This research is based upon work supported by U.S. DARPA ECOLE Program No. #HR00112390060 and U.S. DARPA KAIROS Program No. FA8750-19-2-1004. The views and conclusions contained herein are those of the authors and should not be interpreted as necessarily representing the official policies, either expressed or implied, of DARPA, or the U.S. Government. The U.S. Government is authorized to reproduce and distribute reprints for governmental purposes notwithstanding any copyright annotation therein.

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

# A Robustness to Domain Shift: Zero-shot Cross-Domain Transfer

Table 4: Evaluating robustness to domain shift. We train the models on SSv2-label and perform zero-shot action classification on out-of-domain datasets, i.e., Moments-In-Time [38] and Temporal-Kinetic [45]. $\Delta$ indicates the relative increase/decrease compared to the backbone.

| Method [Patcher Training Loss] | Zero-shot Cross-domain Transfer | | | |
|---|---|---|---|---|
| | Moments-In-Time | | Temporal-Kinetic | |
| | Val (Acc) | $\Delta(\%)$ | Val (Acc) | $\Delta(\%)$ |
| InternVideo Backbone | 23.3 | - | 57.7 | - |
| KP-Transformer FT [VTC] | 16.5 | -29% | 44.7 | -23% |
| KP-Perceiver FT [VTC] | 9.9 | -58% | 24.7 | -57% |
| Side-Tuning [61] [VTC+DVDM] | 21.2 | -10% | 54.5 | -6% |
| PAXION [VTC+DVDM] | 21.6 | -7% | 49.7 | -14% |
| w/o Knowledge Fuser | 4.3 | -82% | 16.3 | -72% |
| w/ Backbone Ensemble | **23.9** | **+3%** | **58.1** | **+1%** |

Humans acquire action knowledge through multisensory interactions, and have the remarkable ability to generalize to new objects and scenarios. Similarly, our ultimate goal is to learn the underlying rules of action knowledge that is generalizable to unseen domains. However, it is highly challenging when we are given only domain-specific datasets. For instance, the SSv2 dataset [13] only has 174 action classes, which is insufficient to capture the full range of open-world actions. The Ego4d dataset is limited to ego-centric videos, making it difficult to generalize to other types of videos. Training on such domain-specific data can easily lead to overfitting to spurious features and introduce catastrophic forgetting of tasks from other domains. In this section, we further explore *whether* PAXION *is robust to domain shift* and *whether the learned action knowledge can bring positive transfer to action-centric tasks on unseen domains*.

We consider a zero-shot cross-domain transfer setting where we directly apply the models trained on SSv2-label [24] to unseen domains. We consider two zero-shot action classification tasks based on **Moments-In-Time** [38][3] and **Temporal-Kinetic** [45]. Moments-In-Time contains 305 action classes with diverse types of videos that are distinct from SSv2, including movie clips, stock footages, and cartoons. Temporal-Kinetic contains 32 manually selected action classes from Kinetic-400, with a special focus on temporal reasoning. We directly use the action labels (e.g., *"bouncing"* and *"kicking"*), as the text candidates for the zero-shot classification [43], which introduces additional domain shifts in terms of text distribution compared with the annotations in SSv2-label (e.g., *"book falling like a rock"*).

**Fusing with the backbone improves robustness to domain shift.** Table 4 shows the zero-shot action classification accuracy and the relative difference $\Delta(\%)$ compared with the frozen backbone. We find that adding the Knowledge Fuser effectively increases robustness to domain shift, as reflected by a smaller negative $\Delta$. The Side-tuning also demonstrate similar benefit via alpha blending between the Knowledge Patcher and the backbone.

**Positive transfer can be achieved by ensembling the Knowledge Fuser (KF) with the backbone.** We further propose a simple inference trick, **Backbone Ensemble**, which combines the output probability from the KF and the backbone model through addition. Specifically, the final prediction of the action class index $c \in 0, 1, ..., C$ is computed as $c = \arg\max_{i \in 0,1,...,C} (p_a(i = c) + p_b(i = c))$, where $C$ is the number of classes, $p_a$ and $p_b$ are the predicted probability distribution from the KF and the backbone respectively. We obtain the final prediction by ranking the combined probability of the action text candidates. Our experiments show that this simple inference technique can effectively enhance zero-shot performance and achieve positive transfer on unseen domains.

# B Details of Action Dynamics Benchmark (ActionBench)

We construct ActionBench based on two existing video-language datasets with fine-grained action text annotation, Ego4d [14] and SSv2 [13]. To automatically generate the antonym text for the Action

---

[3]We subsample 2k instances for doing this evaluation.

Table 5: ActionBench Statistics

| Dataset | #Train | #Eval | Video Type |
|---|---|---|---|
| ActionBench-Ego4d | 274,946 | 34,369 | first-person |
| ActionBench-SSv2 | 162,475 | 23,807 | first-person, third-person |

Antonym task, we leverage WordNet [36][4] to find antonyms for verb text tokens. Additionally, we construct an additional verb-to-antonym mapping by leveraging ChatGPT[5] and manual curation, since the WordNet database does not cover all verbs in the action taxonomy of the dataset. Furthermore, to ensure that the action antonym indeed forms a negative video-text pair with the original video, we exclude verbs that do not have a semantically reasonable antonym, such as "use" and "look". For Ego4d, we consider a subset of EgoClip [32] annotations, for SSv2 we consider the entire dataset. The final statistics of the training and evaluation splits can be found in Table 5. For SSv2, since the test set does not provide label annotation, i.e., annotation with filled object names, we report scores on the validation set. For Ego4d, we evaluate on the test set. For results in Table 1, we train the Knowledge Patcher variants for one epoch on the training sets and report the accuracy on the evaluation sets. We downsampled the videos into 224x224 in scale with a frame rate of 8 fps for both training and evaluation. For human evaluation, we randomly sample 50 instances for the Action Antonym and the Object Replacement task, and another 50 instances for the Video Reversal task. The human evaluation is done by the authors.

## C  Identifying State-change Salient Videos for Action-Temporal Matching (ATM)

As detailed in § 3.1, we formulate the Action-Temporal Matching (ATM) loss as distinguishing reversed video from the original one given an action text. ATM requires the model to learn the correlation between the correct temporal ordering of the visual observations and the corresponding actions. However, some actions, such as "wiping" and "holding", are repetitive or continuous and may not result in visible state-changes across the frames in the video clip. This can introduce additional noise for the ATM loss when the reversed video is indistinguishable from the original one. To address this issue, we propose two metrics to identify state-change salient videos by leveraging image-language foundation models. We use pretrained BLIP [28] to compute (1) **frame-text semantic change** $\delta_{vt}$, which indicates how the frame-text alignment changes across the first half and second half of the video; (2) **frame-frame similarity** $\theta_{vv}$, which indicates how different the frames from the first half and second half of the video are.

$$\delta_{vt} = \left| \frac{1}{N/2} \left( \sum_{i \in [0, N/2)} S(\mathbf{v_i}, \mathbf{t}) - \sum_{j \in [N/2, N)} S(\mathbf{v_j}, \mathbf{t}) \right) \right| \tag{1}$$

$$\theta_{vv} = S \left( \frac{\sum_{i \in [0, N/2)} \mathbf{v_i}}{N/2}, \frac{\sum_{j \in [N/2, N)} \mathbf{v_j}}{N/2} \right) \tag{2}$$

where $N$ is the total number of sampled frames[6], $\mathbf{v}$ and $\mathbf{t}$ are the frame image embedding and the text embedding from pretrained BLIP encoders, $S$ denotes cosine similarity.

Intuitively, if we observe a large frame-text semantic change ($\delta_{vt}$) and a small frame-frame similarity ($\theta_{vv}$), we could expect to see salient state-changes between the first half and the second half frames. We empirically set a threshold for $\delta_{vt}$ and $\theta_{vv}$. During training, we only compute ATM loss on videos that satisfy $\delta_{vt} > 0.003$ and $\theta_{vv} < 0.95$. The metrics are computed off-line thus do not bring computational overhead during training. Figure 7 shows an example of the videos that are kept and skipped based on the computed metrics.

---

[4]We use the WordNet Interface from NLTK https://www.nltk.org/howto/wordnet.html.
[5]https://openai.com/blog/chatgpt.
[6]We use $N = 8$ in our experiments.

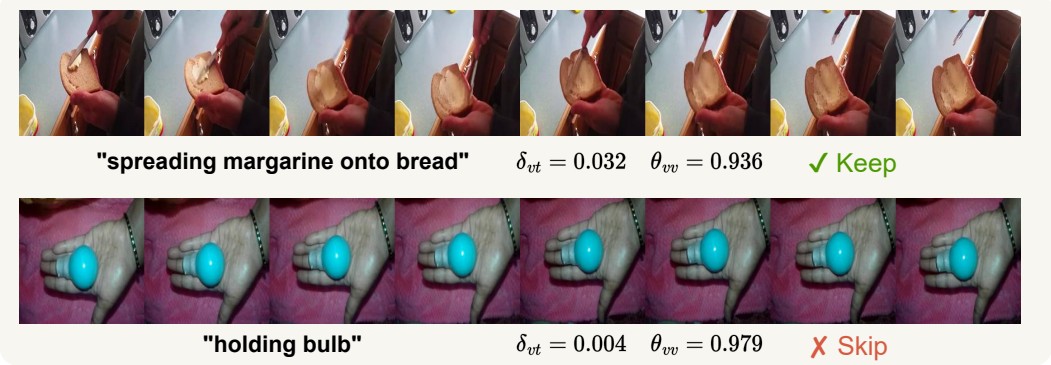

Figure 7: Example of identifying state-change saliency in videos for forward dynamics modeling. $\delta_{vt}$ and $\theta_{vv}$ indicates *frame-text semantic change* and *frame-frame similarity* metrics.

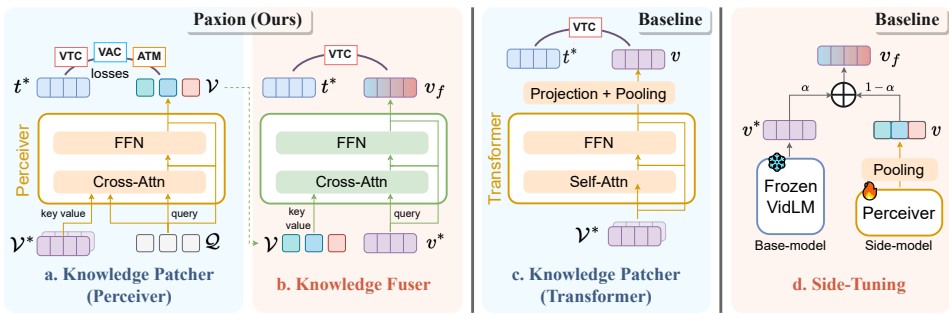

Figure 8: Detailed architecture of Knowledge Patcher (Perceiver), Knowledge Patcher (Transformer), Knowledge Fuser and Side-Tuning fuser.

# D   Implementation Details

## D.1   Architecture Details.

Figure 8 shows detailed architecture of the Knowledge Patcher and Knowledge Fuser in our PAXION framework, as well as the baseline variants being compared in Tables 1, 2 and 3.

**Knowledge Patcher (Perceiver).**   The Perceiver-based Knowledge Patcher contains a single cross-attention layer and a two-layer feedforward network. The Perceiver module performs cross-attention between a sequence of learnable latent queries $\mathcal{Q} \in \mathbb{R}^{l,d}$ and the raw visual embeddings $\mathcal{V}^* \in \mathbb{R}^{P,D}$ from the frozen backbone, where $P$ denotes the visual token length and $D$ represents the hidden dimension of the visual backbone. Since the user-defined sequence length $l$ and hidden dimension $d$ of the learnable latent queries are typically much smaller than $P$ and $D$ from the backbone, the Perceiver module serves as an information bottleneck that extracts knowledge-specific features from the raw visual features. For instance, in the case of InternVideo [52] backbone, we set $l = 16, d = 768$ which is much smaller than $P = 1576, D = 1024$ for each video clip with 8 sampled frames. Similar to BLIP-2 [27], when computing the similarity between the visual tokens $\mathcal{V} \in \mathbb{R}^{l,d}$ from the Knowledge Patcher and the single textual feature vector $t^* \in \mathbb{R}^d$, we first compute the pairwise similarity between each visual token and the text feature vector, and then take a maximum across all visual tokens as the final video-text similarity. The results in Table 1 demonstrate the Perceiver-based Knowledge Patcher achieves competitive or better performance compared to the Transformer variant while being 2-3 times smaller. Additionally, we measure the computation overhead of the two variants, and find that the Perceiver variant requires 10 times fewer *multiply-add operations* than the Transformer variant. This further demonstrate that Perceivers can serve as effective and efficient extractors for knowledge-specific features.

Table 6: Detailed configurations for methods in Tables 2 and 3, and Figure 6.

| Method | has Knowledge Fuser? | Trainable Param# | Patching Objectives | Fusing/Finetuning Objectives |
|---|---|---|---|---|
| KP-Transformer FT | ✗ | 8.4M (1.8%) | VTC | VTC |
| KP-Perceiver FT | ✗ | 4.2M (0.9%) | VTC | VTC |
| Side-Tuning | ✗ | 4.2M (0.9%) | VTC + DVDM | VTC |
| **PAXION** | ✓ | 8.2M (1.7%) | VTC + DVDM | VTC |
| KP+Finetune | ✗ | 4.2M (0.9%) | VTC + DVDM | VTC |
| KP[VTC]+KF | ✓ | 8.2M (1.7%) | VTC | VTC |

Table 7: Detailed training configurations for tasks in Tables 2, 3, and 4.

| Downstream Task | Patching Dataset | Patching #Epochs | Fusing/Finetuning Dataset | Fusing/Finetuning #Epochs |
|---|---|---|---|---|
| SSv2-label [24] | SSv2 | 1 | SSv2 | 1 |
| SSv2-template [24] | SSv2 | 1 | SSv2-template | 2 |
| Temporal-SSv2 [45] | SSv2 | 1 | SSv2-template | 2 |
| NExT-QA [54] | NExT-QA | 1 | NExT-QA | 4 |
| Moments-In-Time [38] | SSv2 | 1 | SSv2 | 1 |
| Temporal-Kinetic [45] | SSv2 | 1 | SSv2 | 1 |

**Knowledge Patcher (Transformer).** The Transformer variant of the Knowledge Patcher is a standard Transformer Encoder which contains a self-attention layer and a feedforward layer. The Transformer Encoder performs self-attention on the raw visual embeddings $\mathcal{V}^* \in \mathbb{R}^{P,D}$ from the frozen backbone and output an updated visual embedding $\mathcal{V} \in \mathbb{R}^{P,D}$. To obtain video-text similarity, we first project the visual embeddings into the same dimension as the textual feature vector $t^* \in \mathbb{R}^d$ and then do mean pooling before computing dot product.

**Knowledge Fuser.** The Knowledge Fuser has the same architecture as the Knowledge Patcher which contains a single cross-attention layer and a two-layer feedforward network. In this case, we use the pooled visual feature from the backbone $\mathfrak{v}^* \in \mathbb{R}^d$ to provide query and the Knowledge Patcher output $\mathcal{V} \in \mathbb{R}^{P,D}$ to provide key and value for the cross-attention. The intuition is to obtain a balanced representation for general downstream tasks by fusing the action-centric KP representation ($\mathcal{V}$) with the object-centric backbone representation.

**Side-Tuning.** As an alternative to the Knowledge Fuser, we consider Side-Tuning [61] for further integrating the Knowledge Patcher with the backbone. Side-Tuning contains a *base-model* and a *side-model*, where the base-model is pretrained and frozen and the side-model is trainable. In our setting, we treat the backbone as the base-model and initialize the side-model using the trained Knowledge Patcher. We then side-tune the Knowledge Patcher along with the backbone using alpha blending. Specifically, the final fused visual feature $\mathfrak{v}_f$ is obtained by $\mathfrak{v}_f = \alpha(\mathfrak{v}^*) + (1-\alpha)\mathfrak{v}$, where $\mathfrak{v}^*$ is the mean-pooled backbone visual feature, and the $\mathfrak{v}$ is the mean-pooled Knowledge Patcher feature. And $\alpha = Sigmoid(a) \in [0,1]$, where $a$ a learnable scalar.

## D.2  Knowledge Patcher Training.

We use two Nvidia Tesla V100 (16GB) GPUs for all experiments. For the Knowledge Patcher variants in Table 1, we train them on the training set of the datasets in the ActionBench for one epoch with either VTC loss only or VTC + DVDM (VAC + ATM) loss. We use AdamW [34] optimizer with a learning rate of 1e-5 and a weight decay of 0.05. For the transformer variant, we use a batch size of 8 per GPU. For the Perceiver variant, we are able to increase the batch size to 32 per GPU due to the reduced computation complexity.

## D.3 Downstream Task Training.

Tables 6 and 7 shows detailed configurations for downstream task training with methods described in Tables 2 and 3, and Figure 6.

As shown in Table 7, the finetuning dataset for SSv2-label is identical to the SSv2 action knowledge patching dataset where the annotations are filled templates, such as "Book falling like a rock". The SSv2-template dataset, on the other hand, contains the object-obscured version of the original SSv2 annotations such as "Something falling like a rock". For the Video-to-Action Retrieval tasks, we consider two different subsets from the SSv2 validation set with the object-obfuscated annotations: SSv2-template [24] and Temporal-SSv2 [45]. SSv2-template contains all 174 action classes while Temporal-SSv2 contains 18 manually selected action classes that require more temporally-demanding distinctions, and cannot be distinguished using shuffled frames, such as "Approaching" and "Moving away". In order to investigate the impact of the action knowledge patching, we do not finetune a dedicated model for the 18 action classes for Temporal-SSv2, but instead use the model trained on SSv2-template to directly evaluate on Temporal-SSv2. Therefore, when observed larger improvements on Temporal-SSv2, we can draw the conclusion that patching with action knowledge contributes more to action-centric tasks (§ 4.2).

The hyperparameters, such as the learning rate, are identical to those used during Knowledge Patching training. For Video-Text Retrieval (SSv2-label) and Video-to-Action Retrieval (SSv2-template, Temporal-SSv2), the DVDM (§ 3.1) objective includes VAC and ATM, while for Causal-Temporal VQA (NExT-QA), we only use VAC. This is because the training instances in NExT-QA are not formatted as video-text pairs but instead are in the format of multiple choice QA, making it not suitable for the ATM loss. Each video corresponds to one question and five candidate answers. We apply VAC to NExT-QA by adding action antonym text for each question as hard negative candidate answers.

For the downstream tasks (in Appendix A) for zero-shot cross-domain transfer (Moments-In-Time [38] and Temporal-Kinetic [45]), we use the model trained on SSv2 to perform zero-shot evaluation.

## E  Additional Qualitative Analysis

Figures 9 and 10 show additional qualitative examples on downstream tasks. The examples in demonstrate that PAXION improves understanding of challenging actions that require fine-grained temporal reasoning on the frames. For example, whether it is ''pretending'' to do something or actually doing that, and whether an object is moving ''towards'' or ''away'' from the camera.

In Figure 11, we show failure cases of PAXION to discuss remaining challenges. We find that PAXION still struggle to understand *negation* and *spatial attributes*. For example, both VTC-Finetune baseline and PAXION fail to distinguish ''without letting it drop down'' from ''then letting it drop down''. For questions that require fine-grained spatial information of objects such as ''how many goats can be spotted'', PAXION cannot perform well. Potential solutions including incorporating the patched VidLM with a code language model to disentangle perception and reasoning similar to ViperGPT [47]. By leveraging the strong logical reasoning ability of a code language model, we can easily solve the negation and counting problems by creating code scripts with booleans and loops, and then use the VidLMs as "API calls".

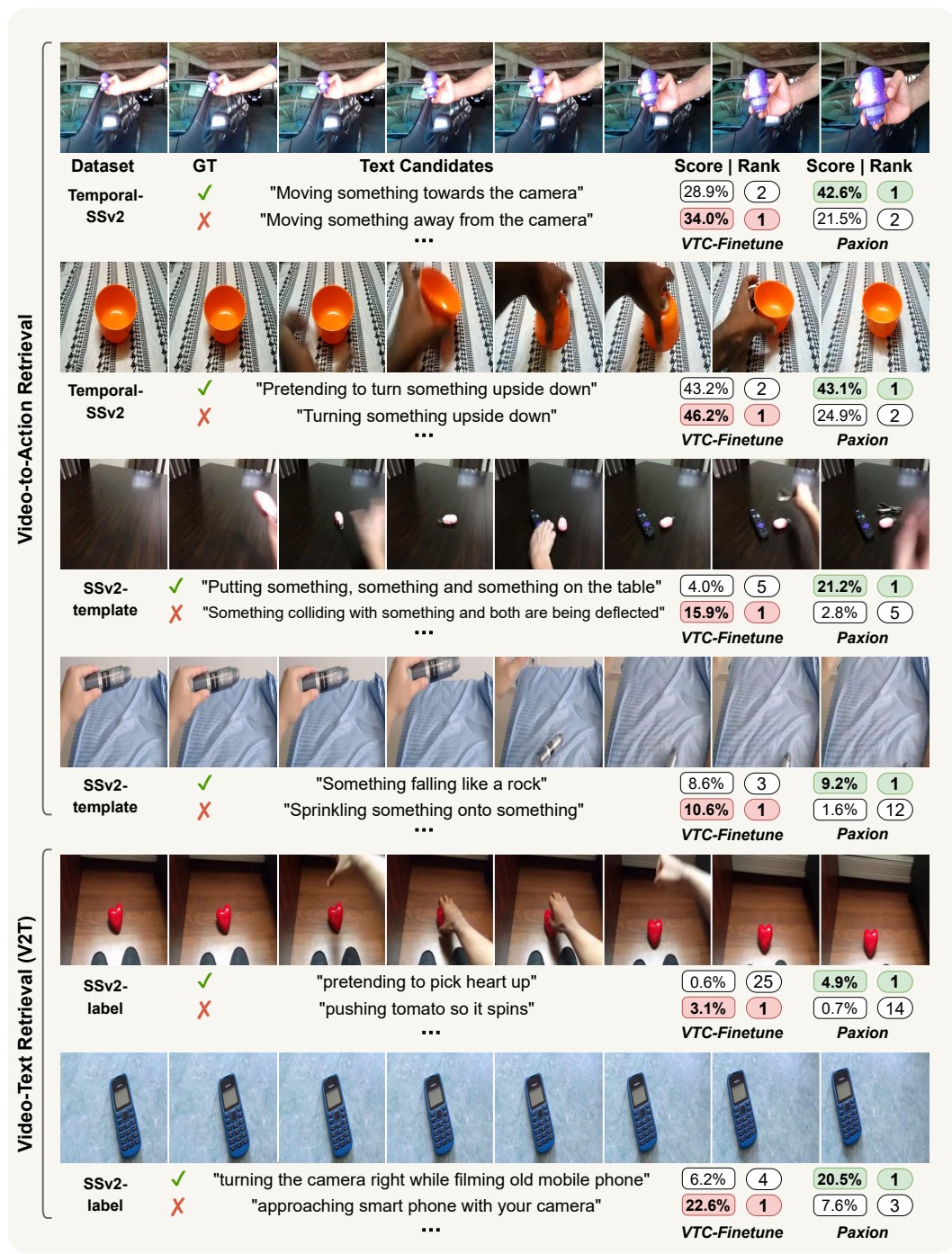

| Dataset | GT | Text Candidates | Score \| Rank | Score \| Rank |
|---|---|---|---|---|
| Temporal-SSv2 | ✓ | "Moving something towards the camera" | 28.9% ② | **42.6%** ① |
| | ✗ | "Moving something away from the camera" | **34.0%** ① | 21.5% ② |
| | | ... | *VTC-Finetune* | *Paxion* |
| Temporal-SSv2 | ✓ | "Pretending to turn something upside down" | 43.2% ② | **43.1%** ① |
| | ✗ | "Turning something upside down" | **46.2%** ① | 24.9% ② |
| | | ... | *VTC-Finetune* | *Paxion* |
| SSv2-template | ✓ | "Putting something, something and something on the table" | 4.0% ⑤ | **21.2%** ① |
| | ✗ | "Something colliding with something and both are being deflected" | **15.9%** ① | 2.8% ⑤ |
| | | ... | *VTC-Finetune* | *Paxion* |
| SSv2-template | ✓ | "Something falling like a rock" | 8.6% ③ | **9.2%** ① |
| | ✗ | "Sprinkling something onto something" | **10.6%** ① | 1.6% ⑫ |
| | | ... | *VTC-Finetune* | *Paxion* |
| SSv2-label | ✓ | "pretending to pick heart up" | 0.6% ㉕ | **4.9%** ① |
| | ✗ | "pushing tomato so it spins" | **3.1%** ① | 0.7% ⑭ |
| | | ... | *VTC-Finetune* | *Paxion* |
| SSv2-label | ✓ | "turning the camera right while filming old mobile phone" | 6.2% ④ | **20.5%** ① |
| | ✗ | "approaching smart phone with your camera" | **22.6%** ① | 7.6% ③ |
| | | ... | *VTC-Finetune* | *Paxion* |

Video-to-Action Retrieval

Video-Text Retrieval (V2T)

Figure 9: Additional qualitative examples (Retrieval).

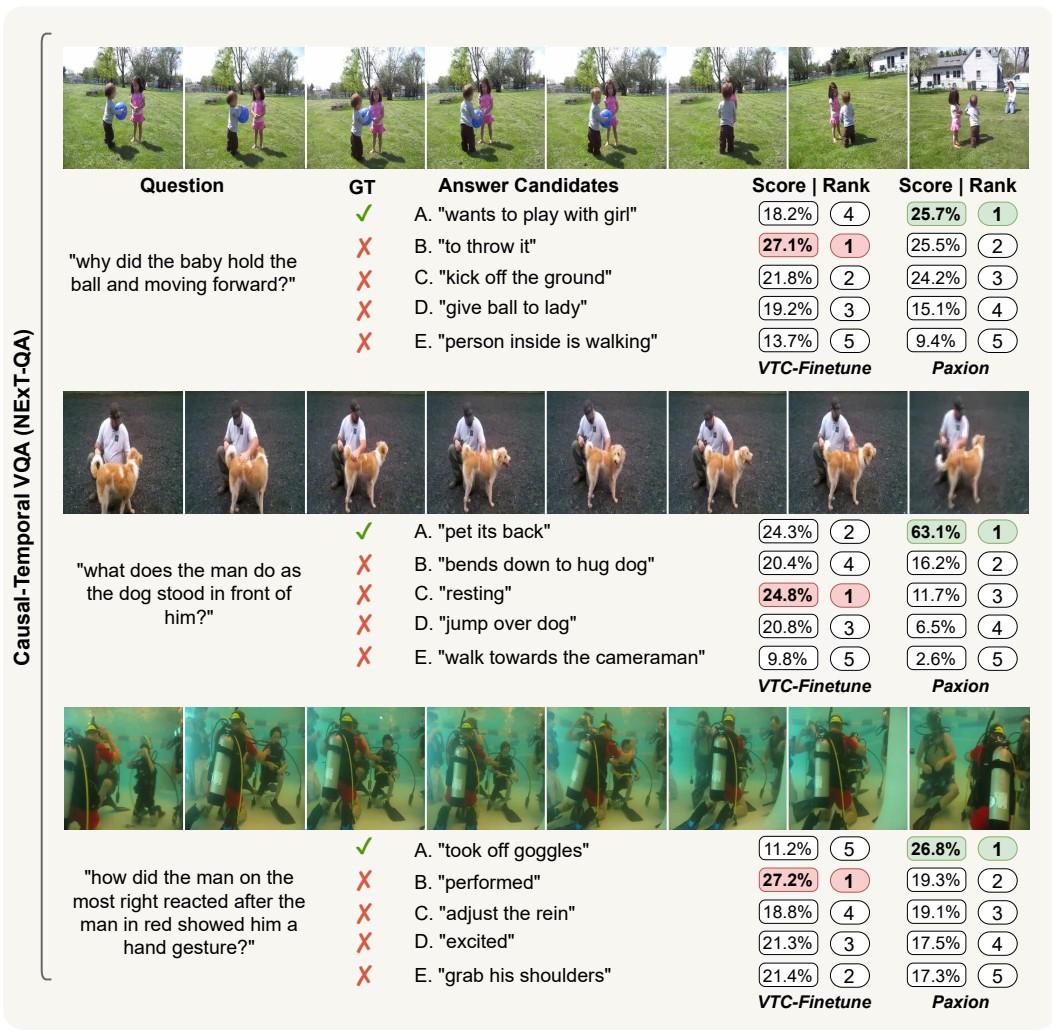

Figure 10: Additional qualitative examples (VQA).

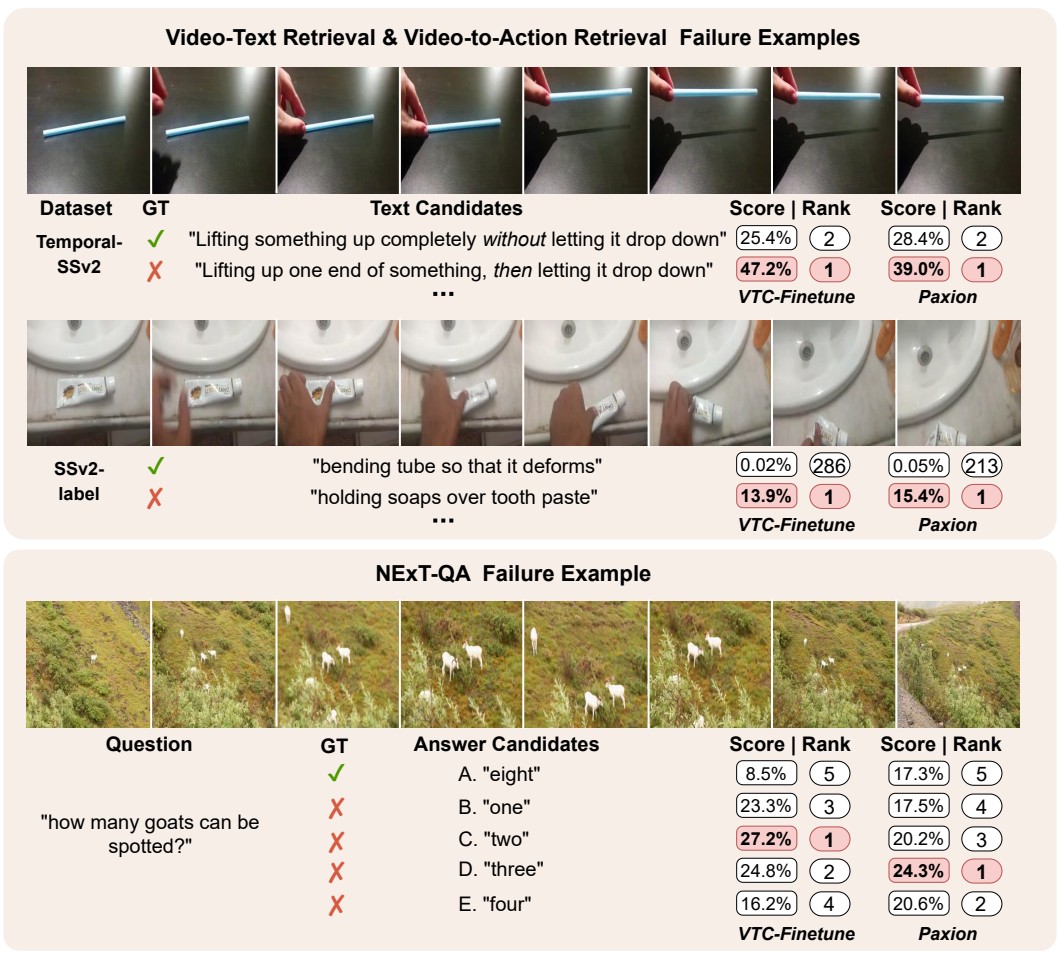

Figure 11: Failure examples.