# OpenReview forum: "Paxion: Patching Action Knowledge in Video-Language Foundation Models"
_NeurIPS.cc/2023/Conference — NeurIPS 2023 spotlight_

### Official Review · Reviewer_BUR1 · 2023-07-04

**Soundness:** 4 excellent
**Presentation:** 3 good
**Contribution:** 3 good
**Rating:** 7
**Confidence:** 5

**Summary:**

The paper tackles the known issue of CLIP-like models acting similar to bag-of-words models, in which structured information (e.g. relations between objects) is not really captured, so for action recognition the information leveraged is mostly object and scene (e.g. a person besides a guitar vs a person playing a guitar). The authors propose to use some extra augmentations related to action recognition information (both on the visual side and the language side), and also add an adaptor (a perceiver/Q-former) that becomes trainable using the extra augmentations. The underlying L&V model is kept frozen, so the training is efficient and the underlying model is not degraded for other domains.

**Strengths:**

- clear issue, sensible yet innovative solution, good results, all in a well-written paper.

**Weaknesses:**


- Right now there is little information in the main text that points to the "patching data", only the appendix has this info. I think this should be made very clear in the text and in the tables. One imagines that there's domain-specific fine-tuning, but it is not a must, it could be that the calibration set comes from a mixture of sources so the model has general applicability off-the-shelf. That the setting is the former is important. Does this make sense?

- The other implication is: this assumes some CLIP-like model for video, and then uses the "knowledge patching" trained on downstream on-domain data. But one could equally decide to go the other way and start with the image CLIP, and train an adaptor to make it work on video using downstream data. This is the approach for example of ST-adapter and AIM. These comparisons are not included in the paper.

(Not important but maybe interesting) The paper (text augmentations) reminded me of the following paper: "CVPR'23, Teaching Structured Vision & Language Concepts to Vision & Language Models", which is not discussed.


**Questions:**

The main question is how this method compares against CLIP + ST-adapter or a similar strategy to adapting CLIP to video through lightweight adapters.

Open to hearing the author's opinion about the first point (the setting)

**Limitations:**

No.
Authors could comment on the setting (need to have downstream training data), or the standard issues with vision + language pre-training for example.

---

> ### Author Rebuttal · Authors · 2023-08-04
>
> We thank Reviewer BUR1 for the constructive comments. We are glad that you find our paper to be innovative and well-written. We will address your comments and questions in the following paragraphs.
>
> ### Little information about the “patching data”
> We appreciate the reviewer’s suggestion to include more details about the setting and the construction of the patching data. In the next version, we will clarify the following points in the main text:
> 1. The construction of the patching dataset (action antonyms and reversed videos, detailed in Appendix B) is fully automatic, meaning that the users can create their own patching datasets easily.
> 2. As shown in Table 7, the patching dataset and the fine-tuning dataset are not necessarily the same (i.e., patching with SSv2-label but fine-tuning on SSv2-template). Hence, as mentioned by the reviewer, it is possible to create a mixture of patching datasets for the model to learn general action knowledge and then apply it to a new task in a zero-shot manner. The zero-shot cross-domain transfer result in Appendix A demonstrates some initial promising results regarding this generalization ability, but we still have a large room for improvements.
>
> ### Comparison with CLIP-adapted models
> We agree with the reviewer that a promising line of work for solving video-language tasks is to initialize and adapt from strong image-language models, such as CLIP. In fact, we also included a recent representative model along this line as one of our backbone, CLIP-ViP. We want to reemphasize that the primary motivation of Paxion is to patch missing action knowledge into existing frozen foundation models, including those CLIP-adapted models. As depicted in Table 1, the original CLIP-ViP still demonstrates nearly random performance on our ActionBench, while Paxion significantly further enhances its action understanding. We thank the reviewer for the suggestion for adding comparison with CLIP+ST-adapter setup and will investigate that in the next version.
>
> ### Comparison with the "CVPR'23" paper
> We appreciate the reviewer's recommendation to compare our work with the paper "CVPR'23, Teaching Structured Vision & Language Concepts to Vision & Language Models." In the next version, we will incorporate the following discussion in the related work section.
>
> Although the high-level idea is related, especially concerning negative instance generation, we identify the following major differences:
> 1. They focus on object attributes and relations in static images, whereas we focus on actions and state-changes in dynamic videos.
> 2. They adopt a LoRA-based PEFT method, while we propose a novel Perceiver-based patch-and-fuse framework, enabling the backbone to be fully frozen.

---

> > ### Comment · Reviewer_BUR1 · 2023-08-18
> > **response to authors**
> >
> > I have read the other reviews and all the author responses. I don't see much to change my original appraisal either in the reviews and the author responses.
> >
> > Regarding the specific reply to my comments:
> > - I would appreciate if the patching data is more explicitly defined. I understood what the authors reply, but was just pointing out that this info is important and it is quite buried.
> > - There is no need to compare against the cvpr paper, nor argue why it is different. I just pointed out a reference that seems relevant and maybe useful. I have nothing to do with the authors of that paper.
> >
> > I'll be maintaining my score, which is already positive.

---

> > > ### Author Response · Authors · 2023-08-18
> > >
> > > Thank you for your tremendous effort in reviewing and providing valuable comments! They are very helpful for enhancing the submission.

---

### Official Review · Reviewer_CJDm · 2023-07-06

**Soundness:** 3 good
**Presentation:** 3 good
**Contribution:** 3 good
**Rating:** 6
**Confidence:** 5

**Summary:**

In this paper, the authors introduce an interesting ActionBench, aiming to handle action antonym and video reversal problems. To remedy the problem in well-trained VidLM, the authors propose the DVDM objective, along with knowledge patcher and fuser. Extensive experiments demonstrate the effectiveness of the novel objective and modules.

**Strengths:**

- Novel and internting ActionBench.
- Simple modified contrastive objective to enhance the VidLM abilities for recognizing action antonym and video reversal problems.
- Extensive ablation studies show the effectiveness.

Overall, I appreciate the paper's idea to handle the interesting ActionBench. Though it has been somehow proposed in SthSth, there isn't any work trying to handle the problem. The authors propose the smart fine-tuning method with the perceiver as a knowledge patcher and fuser. And they design a modified contrastive loss for fine-tuning VidLM.

**Weaknesses:**

In fact, the VAC and ATM losses are modified contrastive losses with different positive and negative examples. However, the authors give a complicated motivation in Section 3.1 for their DVDM, which makes the paper harder to read. I suggest to present their idea in more easy-to-follow way.

**Questions:**

See weakness

**Limitations:**

See weakness

---

> ### Author Rebuttal · Authors · 2023-08-04
>
> We thank Reviewer CJDm for the constructive comments. We appreciate that you find our ActionBench to be novel and interesting. We will address your comments in the following paragraph.
>
> ### Complicated motivation for the DVDM objectives
> We appreciate the reviewer's suggestion, and will revise section 3.1 to provide a clearer presentation of this motivation.
>
> The reason we motivate DVDM from the Markov Decision Process (MDP) is to give a more **generalized formulation** of the new **video dynamic modeling objective**. Due to the unique challenges mentioned in line 171, we leveraged a “relaxed” formulation of the dynamic modeling and integrated it into a contrastive learning framework as VAC and ATM losses. However,  in scenarios with cleaner training data or within a fully controlled environment (e.g., simulator), it is definitely feasible to explore a generative version of the video dynamic modeling objective in future research. Our current "discriminative" version serves as a straightforward yet effective initial step towards fine-grained modeling of visual changes in videos.

---

> > ### Comment · Reviewer_CJDm · 2023-08-21
> > **response to authors**
> >
> > I really appreciate the interesting idea of ActionBench, thus I keep positive for the paper. I hope the author can give a more straightforward introduction for others to follow.

---

> > > ### Author Response · Authors · 2023-08-21
> > >
> > > Thank you so much for your positive feedback and your great effort in reviewing! We appreciate your suggestions on the paper writing!

---

### Official Review · Reviewer_SP6c · 2023-07-06

**Soundness:** 3 good
**Presentation:** 4 excellent
**Contribution:** 3 good
**Rating:** 7
**Confidence:** 4

**Summary:**

The manuscript presents an Action Dynamics Benchmark (ActionBench) with three new evaluation metrics for existing video-language models. Through ActionBench, the authors find that existing video-language models essentially rely on recognizing objects to recognize actions. Hence, a parameter-efficient component named Knowledge Patcher is connected in parallel to the video-language model and is trained with Discriminative Video Dynamics Modeling to improve the action understanding ability of the model. Finally, the authors present a knowledge fuser to infuse the knowledge learned by the Knowledge Patcher into the video-language model for downstream tasks. Empirical results show that training a knowledge patcher for use is effective in improving the video-text retrieval and temporal VQA tasks.


**Strengths:**

- Although it is proposed before that current video models may rely on spatial information to recognize actions in videos, the manuscript is the first to present such a benchmark with proper evaluation metrics to show the problem. This can inspire further work to solve the problem in video understanding.

- The Discriminative Video Dynamics Modeling is straightforward and effective in training the video-language model to be aware of the action and the temporal direction of the videos.

- The approach that the manuscript presents is similar to post-pre-training, and it is shown in Table 2 and Table 3 that such a post-pre-training strategy can benefit downstream retrieval tasks.

- The writing is good and the organization of the manuscript is clear.

**Weaknesses:**

- The evaluation in the downstream task is limited to video-text retrieval on SSV2 and VQA on NExT-QA, which makes it difficult to compare the presented approach with existing methods. Since the evaluation is based on InternVideo backbone, it is possible to provide some other comparisons where InternVideo has some published results.

- Further on the above note, I am not quite familiar with VQA but on SSV2, the performance seems oddly low. Especially for video-to-text retrieval, which is essentially a text-driven classification for SSv2 videos, the performance has reached 61% for EVL-B/16 [1] when 8 frames are used, which also connect a side network to the pre-trained encoder. I didn't find details in both the manuscript and the supplemental material why this is the case. Could you provide an explanation for this?

[1] Frozen CLIP Models are Efficient Video Learners


**Questions:**

- To help object-centric understanding, why don't you use the pre-trained representations from the backbone as well (e.g., concatenate or add them to the KP during fine-tuning)?

**Limitations:**

See weaknesses.

---

> ### Author Rebuttal · Authors · 2023-08-04
>
> We thank Reviewer SP6c for the constructive comments. We are glad that you find our benchmark to be inspiring and our paper to be well-written. In the following paragraphs, we will address your comments and questions.
>
> ### Limited downstream tasks for evaluation
> We completely agree that evaluating more downstream tasks would be ideal. However, as mentioned in the Introduction, many popular video-language downstream tasks still suffer from strong single-frame bias, which does not faithfully reflect a model's understanding of action knowledge. This is precisely why we propose ActionBench; it serves as an initial step in building better Vid-L benchmarks that require true **video** understanding. We will investigate extending the probing tasks to connect with more real-world downstream applications in future work.
>
> ### Low performance on SSv2
> We thank the reviewer for pointing out this interesting observation. Here are several key differences between our setting and previous work. We will make these points clearer and do more investigating in the next version:
> 1. In Paxion, as discussed in Sec 3 and Table 1, both the Knowledge Patcher and the Fuser are much more lightweight (one Perceiver layer; consisting of only 0.9% parameters) compared to the side module in [1]. This may result in limited expressiveness. Nevertheless, we acknowledge that further work on scaling would be beneficial.
> 2. One of the fine-tuning objectives (DVDM) in Paxion emphasizes action understanding, while the video-to-text retrieval (SSv2-label) is more object-centric. Figure 6 shows that DVDM significantly benefits more action-centric and temporal-heavy tasks, such as video-to-action retrieval tasks.
> 3. The backbone model might favor certain datasets. Although InternVideo demonstrates generally better zero-shot video retrieval performance than CLIP on many datasets (e.g., MSR-VTT, MSVD, etc.), there is no result on zero-shot SSv2 in the paper. This makes me wonder if CLIP is favored on SSv2 compared to InternVideo. We thank the reviewer for the insight and will include more controlled experiments to investigate this.
> 4. Currently, we only fine-tuned on SSv2-label for one epoch for efficiency, as our main goal is to demonstrate the impact of the proposed DVDM objective versus VTC. We will investigate fully fine-tuning until convergence in the next revision.

---

> > ### Comment · Reviewer_SP6c · 2023-08-21
> >
> > Thanks for the response. I would encourage the authors to provide some additional results on some popular existing benchmarks to demonstrate a clearer position of the approach in comparison with the existing approaches (ok to include in the supplemental materials too). As for SSv2, I look forward to the full experiment results in the next version.
> >
> > Overall, I keep my positive rating of the manuscript.

---

> > > ### Author Response · Authors · 2023-08-21
> > >
> > > Thank you so much for your revision suggestions and your tremendous effort in reviewing! We are glad to hear that you maintain a positive rating of the submission!

---

### Official Review · Reviewer_ou3W · 2023-07-07

**Soundness:** 3 good
**Presentation:** 3 good
**Contribution:** 3 good
**Rating:** 5
**Confidence:** 4

**Summary:**

This paper addresses the task of improving video-language understanding models, which a particular emphasis on their ability to align described actions to video segments and also their ability to model temporal dynamics. The authors first propose ActionBench, which is a modification of two datasets (Ego4D, SS-v2) for probing the degree to which action antonyms and reversed videos impact the models, and show that frozen VidLMs do not perform these probing tasks well. The authors then propose Paxion + a DVDM training objective: Paxion has a knowledge patcher network (better action encoding) and a knowledge fuser (incorporating into frozen VidLMs), and DVDM extends the standard contrastive VTC loss to better correlate action text with the specific ordering of the video frames. The authors then benchmark their approach on downstream datasets and tasks (SS-v2, NExT-QA, etc.)

**Strengths:**

`+` Action/temporal understanding in large-scale pretrained video-language models is an important topic.

`+` ActionBench contains probing tasks that seem like they may be interesting for future investigations, and the authors plan to release data and code for reproducibility.

`+` The proposed Paxion/DVDM improvements seems to make a difference for the probing tasks relative to the base frozen VidLM, and there are ablations to show this.

`+` Downstream evaluations on both SS-v2 and NExT-QA datasets to characterize the proposed model.

**Weaknesses:**

`-` There seems to be a limitation of how much the proposed techniques/tasks make a difference in the final downstream setting (e.g., side-tuning seems to ~match Paxion, within 0.3 points). This calls into question how over-specific the design of the proposed techniques and objectives are to the particular definitions of the proposed probing tasks in ActionBench (and by extension, how useful those formulations of the probing tasks are in downstream settings beyond SS-v2). (e.g., in [36] there is stronger impact on a larger range of downstream  video distributions/tasks)

`-` The analysis for NExT-QA also seems incomplete. Prior/concurrent work (e.g., [5, 36]) report more detailed breakdowns of the accuracy into the specific causal/temporal splits of NExT-QA (are the improvements coming in the right categories?). Further, these other works report accuracy on a harder subset (ATP-hard) that makes the improvements from their verb/temporal augmentation techniques clearer. There is significant space in the current paper (Table 3) for these numbers (and they do not require additional compute, since they are all subsets), so it is unclear why they are not reported especially since they may help to bolster the core claims of the work.

`-` The probing tasks seem closely related to ones proposed in prior work [5], but there doesn't seem to be a full discussion / comparison in this work of what these tasks add to those other ones (there is a brief citation in related work, but no discussion of this kind). Having a clearer sense of the potential complementarity would help to better clarify the contribution of this work (relatedly, how text/visual biases are controlled for in the probing tasks proposed here).

**Questions:**

Please see the weaknesses section above for questions / areas to address in the rebuttal. Because the preliminary rating is borderline, it will be very helpful to finalize/potentially improve the rating.

**Limitations:**

The authors provide a brief discussion of the limitation on focusing on one type of knowledge patch in the conclusion + dataset cards in the supplement.

---

---

**Post-rebuttal update:**

The authors partially address the weaknesses/comments described in the initial review:

`+` The additional analysis on the NExT-QA results helps to better support their method/claims (compared to the original result, which showed minor differences), and we can see a larger gap on some settings for temporal/causal/etc where it is helping now.

`+` The additional discussion with related work provides helpful context of how these works can be viewed as complementary to one another.

`-` The limitations around the impact to downstream tasks, however, still remain. The authors make the argument that there are not a lot of temporal datasets, which is understandable. However:
1. Given one of the highlighted contributions of the work is proposing a new *benchmark* for probing (ActionBench), it would have been much stronger to make a connection to at least a couple other datasets that represent a spectrum of temporal/action understanding. Does ActionBench performance help to differentiate downstream settings where temporal/action understanding correlates well? Why or why not? (This is something that is illustrated by the proxy tasks in [5] well, in Table 5 of their work; it would have been interesting to see if this benchmark helps to complement/refine that other analysis).
2. Furthermore, the fact that many datasets are not temporal does not mean there aren't *any* additional datasets (or subsets of datasets) for event reasoning that could have been interesting to consider (e.g., [5] considers AGQAv2; there are others like STAR as well; [36] identifies an action-focused split of Kinetics).
3. Finally, it's worth noting the current work considers *two* evaluation datasets (SSv2 and NExT-QA), and SSv2 is also the basis for the original probing task (so that leaves only one fully new dataset to make a connection to). In contrast, related and concurrent work consider *many more* to help establish the broad efficacy of their proposed learning objectives, techniques, and analysis tasks (e.g., [5] considers 5+ downstream, [36] considers at least 4). In general, I don't think it is strictly necessary for papers to benchmark on tons of datasets, but given that a core contribution of the work is centered around a proposed probing task (and corresponding remedy), establishing this connection with at least one other dataset would have significantly improved the contribution of the work.

`-` The author response that "the way we construct the probing datasets...is fully automatic, meaning that it is not confined to a particular dataset such as SSv2" is not necessarily persuasive, since this pipeline is tied to the specific (templated) language distribution of SSv2, and  more importantly, some of the tasks may not make sense on other video distributions besides SSv2 (e.g., reversing the frame order may only make sense on SSv2's short videos of gestures/object interactions, where the reversed video can map to a realistic new video ("push object" $\leftrightarrow$ "pull object"), whereas if you show a model a video of people walking backwards or "uncooking" a pancake, it isn't quite as effective as a probing task since that's not a realistic new action).

---

**Overall (post-rebuttal):** I think the rebuttal response did help to address some key concerns, and I do think there is value in considering this work in the broader context of other related efforts [5, 36], so I am increasing my rating leaning towards acceptance. I hope the points mentioned above regarding continued areas of improvement will be helpful for future revisions and/or later work.

---

---

> ### Author Rebuttal · Authors · 2023-08-04
>
> We thank Reviewer ou3W for the detailed and constructive comments. We appreciate your acknowledgment of the value of our proposed probing tasks and the DVDM objective. We will address your comments and questions in the following paragraphs.
>
> ### Limited impact on downstream tasks
> 1. First, we would like to clarify that the Side-Tuning row is not entirely a standalone baseline since it also incorporates our proposed VTC+DVDM objectives. The motivation for comparing with Side-Tuning is to demonstrate that the Patcher-Fuser framework in Paxion can perform competitively, or even better, than current PEFT frameworks. When comparing Paxion with the baselines that use only the VTC objective in Table 2 and 3, we do observe a significant improvement in Paxion's performance on downstream tasks, highlighting the impact of the DVDM objective. *(Please also see the updated Table 3 below.)*
> 2. As detailed in Appendix B, the way we construct the probing datasets (i.e., the action antonym, video reversal and object replacement) is fully automatic, meaning that it is not confined to a particular dataset such as SSv2.
> 3. We totally agree that it is ideal to show impact on more real-world downstream tasks. Unfortunately, as mentioned in the Introduction, many popular video-language downstream tasks still suffer from strong single-frame bias, underscoring the need for better Vid-L benchmarks. Our ActionBench serves as an initial step in this direction, and we will further explore establishing stronger connections to real-world downstream applications.
>
> ### Detailed analysis of the Next-QA results
> We totally agree with the reviewer's suggestion, as it can bolster the claims made in the paper. We already updated Table 3 to include a comprehensive breakdown of the results (including three types: Causal(C), Temporal(T), and Descriptive(D)), and the results on the ATP-hard split.
> Since the policy does not allow submitting revision, we include the **updated Table 3** in markdown below:
>
> | Method | NExT-QA | | | | | | | |
> | --- | --- | --- | --- | --- | --- | --- | --- | --- |
> | | **Original** | | | | | **ATP-hard**|
> | | C | T | D | all | | C | T | all |
> | InternVideo Backbone | 43.3 | 38.6 | 52.5 | 43.2 | | 27.0 | 27.3 | 27.1 |
> | KP-Transformer FT [VTC] | 46.1 | 45.0 | 61.3 | 48.1 | | 32.5 | 33.6 | 33.0 |
> | KP-Perceiver FT [VTC] | 46.0 | 46.0 | 58.9 | 48.0 | | 30.1 | 31.6 | 30.7 |
> | Side-Tuning [VTC+DVDM] | 54.9 | 52.0 | **69.8** | 56.3 | | 37.4 | 36.0 | 36.8 |
> | **Paxion** [VTC+DVDM] | **56.0** | **53.0** | 68.5 | **57.0** | | **38.8** | **38.1** | **38.5** |
>
> *Table 3: Causal-Temporal VQA (NExT-QA) results (in accuracy %) on the validation set. We consider both the original and ATP-hard~[1] split. We report accuracy for 'all' questions or specific types of questions, including causal ('C'), temporal ('T'), and descriptive ('D') questions.*
>
> As the reviewer expected, the decomposed results show that Paxion helps more on the Causal (‘C’) and Temporal (‘T’) types of questions, and achieves a more significant improvement on the harder subset (ATP-hard) where the temporal and action knowledge is emphasized.
>
>
> ### Detailed comparison with the prior work “Test of Time” [5]
> We identify the following key differences/contribution of our work compared with [5]. We will enrich line 281-283 to make these comparisons more explicit.
>
> 1. While [5] primarily focuses on understanding the temporal ordering between events, our work focuses on general action knowledge, encompassing both causal and temporal understanding of actions.
> 2. We propose very different probing tasks. [5] did not investigate verb or object replacement. The time-order reversal in [5] is reversing the appearing order of two video segments while the Video Reversal task in our ActionBench is reversing the frames of a single video.
> 3. Our proposed Paxion framework enables fast action knowledge patching on frozen VL backbones, while [5] requires post-pretraining the backbone model.

---

> > ### Comment · Reviewer_ou3W · 2023-08-21
> >
> > Thank you for the rebuttal response! I plan to update my final rating/review during the reviewer-AC discussion period, but in the meantime, I want to confirm that this is overall helpful towards some of the key concerns raised in the initial review, and that I don't think I have any further follow-up questions for the authors at this time. In particular, I'm quite glad that the updated NExT-QA subset analysis better contextualizes/confirms that the gains are coming in the most relevant settings (e.g., larger deltas on C/T/hard), and the additional discussion with [5] is also helpful to see. I'm still reflecting over more on some of the other rebuttal arguments on continued limitations/impacts -- at the same time, perhaps it's ok to leave some of this for future work as long as the claims in the paper are well-scoped.

---

> > > ### Author Response · Authors · 2023-08-21
> > >
> > > Thank you so much for your tremendous effort in the reviewing process and your engagement in the discussion. We are glad to see that our response has addressed most of your questions. We really appreciate your insightful suggestions which have made the result analysis much more solid and interesting.

---

### Comment · Area_Chair_t2e9 · 2023-08-20
**Thanks for your response -- AC Comment**

Dear authors,

Thanks for providing your responses to the reviewers' comments. They are comprehensive and answer many of the concerns raised during the initial review phase. While there is still time in the discussion phase, the reviewers can engage for further clarifications.

-- Your AC

---

> ### Author Response · Authors · 2023-08-21
>
> We express our gratitude to all the reviewers for their tremendous effort in reviewing our paper. We also want to thank our AC for their remarkable effort in organizing the author-reviewer discussion. We find these discussions to be incredibly helpful and constructive.

---

### Decision · Program_Chairs · 2023-09-21

**Decision:**

Accept (spotlight)

**Comment:**

The paper received primarily positive reviews from 4 reviewers. Several questions were raised during the review period, including about performance analysis, missing comparisons with relevant prior work, and choice and generalization to downstream tasks made in the paper. The rebuttal seems to have addressed many of these concerns effectively. The authors are strongly encouraged to incorporate the information from the rebuttal and subsequent discussion into the final version for completeness.